# Towards a Sustainable Nutrition Paradigm in Physique Sport: A Narrative Review

**DOI:** 10.3390/sports7070172

**Published:** 2019-07-16

**Authors:** Eric R. Helms, Katarina Prnjak, Jake Linardon

**Affiliations:** 1Sports Performance Research Institute New Zealand (SPRINZ), Faculty of Health and Environmental Science, Auckland University of Technology, Private Bag 92006, Auckland 1142, New Zealand; 2School of Psychology, Deakin University, 1 Gheringhap Street, Geelong, VIC 3220, Australia

**Keywords:** bodybuilding, physique, psychology, binge-eating, body image, eating disorder, nutrition, muscle hypertrophy, body composition

## Abstract

Physique athletes strive for low body fat with high lean mass and have higher body image and eating disorder rates than the general population, and even other weightlifting populations. Whether athletes with a background or tendency to develop these issues are drawn to the sport, or whether it drives these higher incidences, is unknown. However, the biological drive of cyclical energy restriction may contribute to binge-eating behavior. Additionally, requisite monitoring, manipulation, comparison, and judgement of one’s physique may contribute to body image concerns. Contest preparation necessitates manipulating body composition through energy restriction and increased expenditure, requiring dietary restraint and nutrition, exercise, and physique assessment. Thus, competitors are at mental health risk due to (1) pre-existing or predispositions to develop body image or eating disorders; (2) biological effects of energy restriction on eating psychology; and (3) dietary restraint attitudes and resultant physique, exercise, and nutrition monitoring behavior. In our narrative review we cover each factor, concluding with tentative best-practice recommendations, including dietary flexibility, slower weight loss, structured monitoring, gradual returns to offseason energy intakes, internal eating cues, appropriate offseason body compositions, and support from nutrition and mental health professionals. A mental health focus is a needed paradigm shift in bodybuilding nutrition practice and research.

## 1. Introduction

Physique sport, or “competitive bodybuilding”, has existed since the turn of the century; however, judging standards and the philosophical ethos of its actors have changed over time. One of the first recorded physique competitions was attached to a weightlifting meet in 1898 in England [1]. This was a popular format well into the 1900’s and even the Mr. America competition—which began in 1939—was held in conjunction with the Senior Weightlifting National Championships during its initial decades. Mr. America contestants received points not only for their physique development, but also for lifting weights, their athletic background, and character traits. Rather than emphasizing extreme muscularity, points were given to contestants with physiques that were muscularly balanced and symmetrical. Indeed, there were separate “most muscular” and muscle-group specific awards given in addition to the standard placement awards [2]. The first pure physique competitions were regional qualifiers in the three years prior to the “The Great Competition” of 1901 held in London. While this was an event to determine the “best built man” in England, winners were selected based on the most aesthetic and balanced physique, as the judges believed form followed function and such a physique represented health and athleticism [1]. Artistic expression was also considered, as the event included Arthur Conan Doyle, physician and author of Sherlock Holmes, and Charles Lawes, amateur athlete and sculptor, as judges alongside strongman-performer Eugen Sandow, the organizer and so-called “father of modern bodybuilding” [2].

However, over time, physique sport has become more and more disassociated from holistic notions of performance, art, or health. In the modern era, competitors in the bodybuilding division are judged purely on the appearance of muscular size, proportions, and definition, and while competitors in other divisions (men’s physique, figure, and bikini) still must retain a primary emphasis on low body fat and high muscular definition, there is additional emphasis placed on attractiveness and presentation [3]. As the criteria have become more extreme, demands placed on athletes have increased as well. To achieve these outcomes, modern competitors engage in resistance training to develop as much muscle mass as possible (during the “off-season”), typically training for years before first dieting to participate in a contest (with subsequent off-seasons between competition diets). Contest preparation diets (the “in-season”) often last months to achieve the combination of maximum muscularity with minimal body fat [4], in some cases with only essential levels of body fat remaining among elite male [5] and female competitors [6].

While evidence-based guidelines for contest preparation provide recommendations on nutrition and training to optimize body composition and minimize deleterious effects on health [4,7], reviewers also report high rates of mental health problems among competitors [4]. Further, the reviewers note insufficient research exists to determine whether biological effects of semi-starvation, pre-existing psychopathology, or social stress specific to competition preparation drive potential harm [4]—or likely, some combination of all three. Finally, other data indicate potential harm may stem from how dietary [8], body weight [9], and physique monitoring [10] occur during energy restriction, and due to the disposition towards dietary restraint [11].

While no modern, quantitative data regarding body image and eating disorders in large cohorts of physique competitors exist to the authors’ knowledge, small cohort studies over multiple decades reported high rates of eating and body image disorders and symptoms among physique athletes. Specific examples are outlined in Section 2; but, to initially familiarize the reader with their magnitude, we present the example of a 1991 cross sectional comparison of female physique athletes and weightlifting controls [12] in which physique athletes had significantly (*p* < 0.05) higher rates of uncontrollable urges to eat (58%), obsession with food (67%), and expressed greater terror of becoming fat (58%). Further, among men (90% of sample), higher body image concerns in bodybuilders compared to weightlifting non-competitors was reported in a recent meta-analysis [13], with symptoms associated with anxiety, depression, neuroticism, and perfectionism. With that said, it is unclear whether competitors see their behavior, cognition, and emotional status as problems requiring mental health intervention, or simply required aspects of their chosen sport. However, in some cross-sectional studies of both male [14] and female competitors [12], a substantial portion were previously diagnosed with a body image or eating disorder (31.25% with binge eating disorder or body dysmorphia [14] and 42% with anorexia [12], respectively), indicating at least at one point, many of these athletes sought help regarding their relationship with their body and/or food.

The trait predispositions among competitors, the biological effects of dieting, and the monitoring process of contest preparation are each a potential driver of harm, and therefore a topic of discussion, as shown in Figure 1. Therefore, this narrative review explores: (1) the psychological profiles and tendencies of physique competitors; (2) the biopsychosocial demands of contest preparation; (3) the nutrition and physique manipulation practices of physique competitors; and 4) the potential pros and cons of various physique and nutrition monitoring strategies to 5) inform future research and tentative guidelines for coaches and competitors to minimize the risks associated with developing psychopathology and psychosocial distress.

## 2. Psychological Profiles and Tendencies

In order to maximize their chances of obtaining high-level performance during competition, bodybuilders—like many other athletes—employ nutritional strategies that are aligned with the requirements of their sport [15,16]. However, the main training goals in bodybuilding are centered on physical appearance; specifically, the aim to attain muscular, symmetrical, and lean appearances to be successful in competition [17]. Due to these requirements, bodybuilders sometimes are strict in the pursuit of their goals which, although physically rewarding, can lead to an adverse relationship with food, training, and body image. Many scientists [18,19,20] are interested in the psychological profile of bodybuilders, by primarily comparing the profile of a body builder to the profile of other athletes or to the general population.

Regarding male bodybuilders, while they appear to be more satisfied with their physique than males in other sports [21,22], they also exhibit more muscle dysmorphia symptoms [23]. Muscle dysmorphia is a body image disorder comprised of a core belief and fear that one is not sufficiently muscular [24]. People who exhibit muscle dysmorphia experience significant distress when their bodies are seen in public, have impaired social and occupational functioning, and are more likely to engage in anabolic steroid use [24]. While there is nothing inherently wrong with the goal of increasing muscle mass to levels beyond what is considered “normal” for competitive purposes, coaches and athletes should ensure that this pursuit does not become pathological and impact a bodybuilder’s sense of self-worth, esteem, or identity outside of the competitive confines of his or her sport (specific guidelines for coaches are present at the end of this review). Longobardi et al. [25] reported more interpersonal problems, obsessive-compulsive tendencies, anxiety, depression, and lower self-esteem among male bodybuilders with symptoms of muscle dysmorphia relative to asymptomatic bodybuilders, especially among younger-age groups. Similarly, lower body satisfaction and greater obsessive behavior was observed among novice bodybuilders when compared to recreational gym users [26]. Moreover, a combination of muscle dysmorphic and obsessive-compulsive symptoms can be observed in individuals who have excessive exercising tendencies [27]. Smith and Hale [28] constructed a specific term “bodybuilding dependence” to describe a set of behaviors similar to exercise dependence, but specific for a bodybuilding population. Emini and Bond [29] observed a significant relationship between bodybuilding dependence and negative outcomes, such as anger, hostility, aggression, and perceived stress level, and they identified mood control as the strongest motive for engaging in the mentioned behaviors. Lastly, neurophysiological observations (alongside self-report data) among bodybuilders and non-exercising men indicate excessive bodybuilding could be seen as a special form of muscle dysmorphia [30], which additionally highlights the severity of consequences related to a compulsive, extreme relationship with bodybuilding.

It is important to consider the personality of bodybuilders when predicting their sport performance and/or mental health [18], as there is evidence that certain personality traits in this population might place someone at a higher risk for pathological behaviors, or that some individuals with pre-existing psychopathology are attracted to bodybuilding [31]. For example, bodybuilders (especially those who use anabolic steroids) reported high perfectionism and feelings of ineffectiveness, but low self-esteem and interoceptive awareness [32]. Pawłowska et al. [20] observed that bodybuilders in their study had greater needs for dominance, higher impulsiveness, aggression, and competitiveness, but lower tolerance for frustration relative to a control group of non-exercising men. Moreover, in research by Bjørnestad et al. [33] competitive bodybuilders experienced higher social isolation and perceived negative stereotyping from others regarding their lifestyle. Further, it seems that earlier experiences of childhood bullying could be associated with muscle dysmorphia in adulthood, which is a relatively common report among bodybuilders [34]. On average compared to controls, bodybuilders may experience more issues with social functioning, while also having low self-esteem and perfectionistic tendencies, which is a set of features commonly observed among individuals with subthreshold [35] and threshold eating disorders [36].

With regards to pre-existing psychopathology, Pope et al. [37] observed that 3% of male bodybuilders reported a history of anorexia nervosa (AN) diagnosis (~100 times the rate of occurrence in the general population at the time of publication in 1993), which is in accordance with findings that being underweight as a young male adolescent has a negative impact on body image, self-esteem, and social adjustment in adulthood [38]. Regarding female competitors, when a sample of bodybuilders was compared with AN patients, Tolle et al. [39] observed certain physiological similarities between the two groups; specifically, both experienced low levels of leptin, T3, and estradiol. Likewise, Davis and Scott-Robertson [19] concluded that bodybuilders on average share some similar psychological traits as AN patients, i.e., high levels of obsessiveness (to a similar degree as patients with obsessive-compulsive disorder), perfectionism, anhedonia, and pathological narcissism. However, the authors addressed an important difference between the two groups and that is the perception of self-worth that seems to be healthier and more positive among bodybuilders [19]. With regards to bulimia nervosa (BN) and binge eating disorder (BED), male bodybuilders (especially recreational) have less symptoms than males with BN diagnosis [40]. Mangweth et al. [41] also noted that, though both groups had obsessional preoccupations with their body image, eating disorder patients were “fat-phobic” and bodybuilders rather experienced a fear of not being muscular enough. Hence, despite both populations overlapping in some traits and behavior, it seems the underlying motivational factors might be less detrimental in the case of bodybuilders, which potentially leads them to more adaptive outcomes.

The fear of insufficient muscularity also follows female bodybuilders [42]. Indeed, Peters and Phelps [43] observed that on average female bodybuilders set increasingly higher standards and goals in terms of muscularity and perceive themselves as less muscular than others perceive them. This is an indication of the body image distortion following an insufficient fitting in stereotypical appearance, in this case, that of being muscular [43], which could subsequently facilitate body image issues [44]. Furthermore, on average female bodybuilders may exhibit higher levels of eating disorder psychopathology than male bodybuilders [45], which at least in non-exercising populations is a robust finding [46]. More specifically, according to research by Walberg and Johnston in 1991 [12], 42% of female bodybuilders used to have a diagnosis of AN (compared to 1% of lifetime prevalence found among women in general; [47]), 67% reported being terrified of becoming fat, and 50% of the sample had uncontrollable urges to eat. In addition, 25% of female bodybuilders reported having abnormal menstrual cycles in a study by Kleiner et al. [48]. With this said, more recent research is needed to confirm if these high rates of pre-existing psychopathology are present in modern day bodybuilders and if these rates differ among those using and those not using anabolic steroids. Female competitors, on the whole, may have higher rates of eating and body image concerns than males; yet, it is unknown if this is a reflection of more women with an existing clinical, sub-clinical, or tendency towards these psychopathologies being drawn to the sport, or if bodybuilding plays some causative role. Nonetheless, female bodybuilders may benefit from being especially cautious when deciding to participate in competitive physique sport. We recommend they adopt evidence-based practices that take into account mental health and consider guidance from appropriate mental health professionals experienced with sport and exercise psychology and eating disorder pathology, to ensure good physical and mental health whilst competing. With that said, the bodybuilding professional coaching community should also be aware of the greater likelihood that competitors have previous experiences of psychopathology related to food and body image, among both female and male athletes, which could be exacerbated by the competition process.

## 3. Biological and Psychological Effects of Contest Preparation

The final phase prior to a physique competition, after sometimes years of intentional muscle building in the “offseason”, is a lengthy contest preparation phase whereby substantial fat loss is achieved by the lowering of energy intake and increase of energy expenditure through greater exercise volumes (e.g., additional aerobic training, higher number of repetitions per set, etc.) [4]. It has been observed that in drug-free “natural” bodybuilders, a contest preparation leads to certain physiological benefits, such as lower blood pressure, resting heart rate, wave reflection, and arterial stiffness [49], as well as some psychological benefits, including a sense of empowerment to undertake new challenges [50]. However, a prolonged decrease in energy availability that occurs as a result of the requisite period of reduced energy can also result in negative adaptations related to the downregulation of certain physiological processes (Table 1) as the body attempts to maintain energy homeostasis [51], particularly resting metabolic rate and other components of total energy expenditure decrease, cortisol increases, and testosterone decreases, among other adaptations [52]. Among female competitors, altered hormonal levels (estradiol and testosterone) lead to menstrual irregularities [53], which is commonly observed among female bodybuilders [48,54]. Generally, female athletes are at a risk for Relative Energy Deficiency in Sport (RED-S) that usually follows a low energy availability state, usually resulting in deteriorations of metabolic rate, menstrual functioning, and impaired bone and cardiovascular health [55]. Halliday et al. [51] found energy availability to be below the threshold (30 kcal/kg FFM/day) throughout contest preparation in one female competitor who did not retain normal menstrual cycle function for a year and a half post competition. In addition, Helms and colleagues [4] contend some “peaking” strategies occurring in the final week and days of preparation, including excessive dehydration and electrolyte manipulation, may be unnecessary with neutral impacts on appearance at best, safe with a negative impact on the physique in some cases, and at worst, in extreme cases, potentially life threatening.

Beyond potential debilitating effects on physical health, some impairments caused by this alteration of eating habits can also manifest in one’s emotional state and behavior. For example, in one case study a seven-fold increase in mood disturbance was observed among a professional natural bodybuilder in the final stages of competition preparation [5]. Moreover, Nindl et al. [56] exposed young, healthy men to an energy deprivation period, after which they reported fatigue, diarrhea, and issues with sleeping, but also an increased urge to consume fatty and sweet-tasting food. In order to avoid such food cravings, physique competitors sometimes begin to avoid social situations [50]. Hence, it is not surprising that Goldfield, Blouin, and Woodside [40] reported higher levels of binge eating and BN symptoms among male competing bodybuilders compared to recreational bodybuilders. Moreover, as a form of monitoring their ongoing competition readiness, bodybuilders frequently check and scrutinize their appearance, particularly their muscularity levels [13], which is strongly linked with muscle dysmorphia symptoms [57] and eating disorder symptoms [10]. Perhaps not surprisingly, the negative psychological consequences of physique competitors are mostly centered around food, eating, and body image.

The food restricting behaviors physique competitors usually exhibit arguably have an effect on their post-competition psychological wellbeing as well. Rossow et al. [5] observed mood disturbances returning to baseline in a competitor six months after competition and a three- to four-month period post-competition was required for most hormone levels to return to normal among a cohort of female competitors in a study by Hulmi et al. [53]. In general, females may potentially experience greater struggles with gaining weight during these eating transitions, as they often dislike their off-season body appearance [45]. However, though the effects of competition participation diminish after a certain time period in most studies, these effects should not be dismissed as inconsequential or unworthy of note. For example, 81% of a bodybuilding sample in a study by Andersen et al. [58] reported reinforced food preoccupations and binge eating, as well as increased anxiety and anger after competition. Generally speaking, some researchers [59,60] speculate that binge-eating could develop as a reflection of hyperphagia (an abnormally high desire for food) that usually occurs after significant weight loss. This is in accordance with findings by Saarni et al. [61] who observed that frequent weight cycling (due to competitions) may predispose a bodybuilder for health problems, such as obesity. These findings provide support for the “dietary restraint theory” which proposes that strict dieting promotes binge eating through both physiological (i.e., reduced sensitivity to hunger and satiety cues, prolonged hunger) and psychological (i.e., perceived deprivation, strong urges to eat, black-and-white thinking styles) mechanisms, which then encourages further (and more inflexible) dietary practices, resulting in a vicious cycle [62].

It is important to point out that there are numerous factors other than dietary restraint that may potentially induce binge eating-like behavior. For example, elevated symptoms of depression and negative affect could be associated with binge eating, as suggested by Goldfield, Blouin, and Woodside [40], who found that competitive bodybuilders reported more frequent binge eating than recreational bodybuilders, despite both groups reporting similar levels of dietary restraint. This finding is consistent with the proposal that negative affect triggers binge eating [63], particularly in people with elevated levels of dietary restraint, because of the belief that eating provides comfort and distraction from any negative emotion experienced. Moreover, other researchers argue that binge eating and other unhealthy behaviors could be exacerbated by bodybuilders’ who judge their self-worth largely, or exclusively, on their ability to control their weight and shape [64]. Finally, Daw and Loxton [65] examined impulsivity underlying binge eating and concluded that reward sensitivity could reinforce attention towards food-related cues and consequently prompt cravings, which is linked to the onset of binge eating behavior. Nevertheless, the evidence for the role of personality in post-competition binge eating development is in its infancy, but this gap in knowledge warrants more research that will clarify the effects of physique competition engagement on one’s mental health. Suffolk [12] warned about the potentially negative impact that some misleading research findings might have on the image of bodybuilding as a health-beneficial activity from a long-term perspective. All in all, it is important to keep in mind that the vast majority of studies in this area are cross-sectional; therefore, conclusions derived from only short-term observations should be taken with caution. While it is possible that certain behavioral tendencies emerge as a result of contest preparation, it is also a likely explanation that individuals with specific sets of characteristics are driven to participation in this sport.

## 4. Nutrition and Physique Manipulation Practices of Bodybuilders and Physique Competitors

Since the 1980’s [66], physique competitors have been studied to determine what strategies they employ to reach the extreme body composition outcomes demanded by competition. As an obscure sport, practices may vary significantly from individual to individual as they appear to be impacted by region [67], sex [68,69], and competitive division [14]. Additionally, nutritional practices of competitors may have evolved over time [66]. While some data indicates competitors on the whole do not engage with evidence-based information [14,66,70], one interview specifically with “natural” bodybuilders who compete in drug-tested events suggested that at least in some cases they engage with, and are guided by, scientific research [71]. However, among competitors in untested federations using anabolic steroids and performance enhancing drug polypharmacy, some reports indicate that information may be primarily obtained from other members of this niche community [70,72,73]. All in all, it seems there are sub-groups within the broader bodybuilding community, each of which places less or more emphasis on scientific data and evidence-based practice.

When assessing literature of prior decades, authors report escalations of dietary restriction occur via progressive energy reduction [66,74,75] and diminishing food variety [68,76,77]. While similar findings regarding food variety reduction exist in some recent research [14,70], contrasting data exist showing that while energy intake progressively decreases during contest preparation—perhaps obligatorily to achieve the required leanness—food variety does not necessarily narrow among some modern physique athletes [67]. Competitors followed bland, monotonous, and repetitive diets in prior decades [68], sometimes to the point where micronutrient shortfalls existed [4]. Authors of this era advised bodybuilders to use the exchange system to increase diet variety while maintaining macronutrient distribution [76]. This advice may have come to fruition, as in a number of recently published case studies of natural physique competitors it appears athletes self-select foods to reach certain macronutrient targets or are guided to do so by their coaches [5,78,79]. Indeed, many competitors adopt an approach known as “if it fits your macros” (IIFYM), in which meal plans are constructed without food source restriction to reach specific targets for grams of protein, carbohydrate, and fat on a day-to-day basis. This approach is in contrast to meal plans constructed from limited lists of preordained “bodybuilding appropriate” foods for each macronutrient category [80].

The IIFYM approach has begun to generate some research attention. For example, the authors of a recent study compared macronutrient-based dieting bodybuilders following an IIFYM approach to “strict-dieting bodybuilders” following a rigid meal-plan approach. The authors reported no significant differences between the two approaches in nutrient quality among males; however, lower energy intakes and more pronounced micronutrient deficiencies were observed among the “strict-dieting” sample of females [80]. For some, the IIFYM approach may result in a more inclusive diet and it also may avoid some aspects of dietary restraint which could put competitors at a higher risk of developing disordered eating. For example, the IIFYM approach is more consistent with flexible dietary control (i.e., a graded approach to dieting) than rigid dietary control (i.e., an all-or-none approach to dieting) which have both been related to disordered eating, weight loss maintenance, and mood disturbances in opposite directions [81,82,83].

However, while IIFYM shares the similarity of not viewing foods in a binary “good or bad” manner with flexible restraint, IIFYM does not necessarily preclude obsessive tracking behavior or all rigid attitudes. For example, competitors may weigh foods to the exact gram or hit a specific target for protein, carbohydrate or fat, to the gram, on a day-to-day basis with minimal deviation [71]. Further, like those with a rigid-restraint mindset being “on or off” a meal plan-based diet, competitors may view macronutrient targets in similar black and white terms. Regardless of whether meal plans based around eating specific foods at specific times in specific amounts are implemented or whether specific targets for protein, carbohydrate, and fat are used to construct diets, the attitude towards restraint [11] and the process of tracking [8,84] are potential factors which could possibly exacerbate disordered eating symptoms in some. Additionally, monitoring one’s progress, via regular self-weighing [85], visual assessment, or comparison-making [10,86], could also play a role in the many mental health problems experienced in this population.

To summarize, approaches to nutrition tracking and monitoring physique progress may have evolved over time [66]. Additionally, there are paradigms which differ in their rigidity within the physique community [80]. However, no research yet exists which examines how these approaches impact mental health, competitive longevity, and athlete fulfillment, or whether they could be modified to improve these outcomes or reduce potential harm. At best, given higher rates of body image dissatisfaction and disordered eating are present among competitors than in the general weight training population [12,13,40,41,64], strategies and approaches used in clinical settings for individuals suffering from body image and eating disturbances should be considered as potential pathways to mitigate harm to competitors and increase competitive career sustainability.

## 5. Nutritional Approaches for Minimizing Harm

Since numerous different dietary and weight-control behaviors are practiced by competitive bodybuilders, it is important to critically review the existing evidence surrounding the potential benefits and harms of these behaviors, as this could inform the development of future “best practice” guidelines for this sport. In this section, we review the literature on common dietary and weight-control practices employed across a variety of different populations (e.g., general, clinical population, and athletic populations), and discuss the potential benefits and harms associated with these practices. While many of the findings reviewed in this section are based on studies sampling general and clinical populations, the fact that many individuals (~30%) from these populations also adopt similar dietary and weight-control practices (like competitive athletes) to attain a certain physique suggests that these data may be loosely generalized to competitive athletes. Moreover, existing approaches to treating and preventing disordered eating symptoms and body image concerns are not diagnosis-specific, meaning that anyone (regardless of whether they have a diagnosis of an eating disorder) may be able to benefit from these approaches, so long as they exhibit these symptoms [87]. This is particularly true of competitive athletes, who are known to exhibit higher rates of disordered eating, eating disorders, and body image concerns than the general population [12]. However, we recognize the need for more research in this area to be conducted exclusively on athletic populations and so our recommendations and guidelines presented throughout this paper are tentative. Further, as an important reminder, it is not established whether physique competition plays a causative role in body and image disorders, or whether those with them or with a higher risk of developing them participate; this “chicken or egg” scenario requires further research to delineate, but the mitigation of harm in such cases is still of paramount importance. Readers must take this into account.

### 5.1. Dietary Restraint: Distinguishing between Rigid and Flexible Control

Success as a bodybuilder depends on, at least in part, the ability to successfully restrain eating. However, as previously discussed, dietary restraint is not a unitary construct [83]; it can be usefully divided into two components: rigid control and flexible control. Rigid dietary control behaviors include disciplined energy intake counting, eating only diet foods to prevent weight gain, avoiding desired energy-dense foods, and fasting or skipping meals for weight-related purposes. This approach stands in contrast to flexible dietary control, which is defined by behaviors such as eating a wide variety of foods while still paying attention to one’s weight, taking smaller serving sizes than desired, and compensating at later meals (consuming “healthier” foods) if “unhealthier” foods were consumed earlier [83]. In that regard, a flexible approach to dieting is considered to be more adaptive and sustainable than a rigid approach [88].

A solid body of research indicates that a rigid approach to dieting is associated with adverse outcomes in both males and females. For example, fasting for long periods of time (a rigid control behavior) has been shown to prospectively predict binge eating and related behaviors in adolescent girls [89], and in women with BN and BED [90]. Moreover, cross-sectional research examining the impact of a specific rigid control behavior, namely meal-skipping, has linked this behavior with an increased frequency of binge eating in women with BED [91] and with depressive symptoms, anxiety symptoms, and quality of life impairment in women with AN and BN [92]. Finally, numerous other cross-sectional studies using the Rigid Control subscale of the Cognitive Restraint Scale—which assesses the broad range of inflexible dietary behaviors [83]—have reported consistent and robust links between rigid control and numerous adverse health outcomes in both male and female participants, including disordered eating behaviors and attitudes (e.g., binge eating, disinhibited eating, dichotomous thinking), body image concerns (e.g., shape/weight overvaluation, body dissatisfaction), psychological distress (e.g., depressive and anxiety symptoms), and poorer wellbeing [11,83,93,94,95,96]. Taken together, the available evidence suggests that a rigid approach to dieting may be potentially detrimental.

On the other hand, there is evidence to suggest that a flexible dietary approach may be healthier to adopt than a rigid approach. For example, several cross-sectional studies of student and general populations have reported associations between flexible dietary control and positive health outcomes, including lower levels of disordered eating, body image concerns, body weight, and psychological distress [83,88,94,97]. Moreover, in people with BED, increases in flexible dietary control during the course of cognitive-behavioral therapy were associated with binge eating abstinence and greater percent weight loss [98]. Finally, Teixeira et al. [99] found that increases in flexible control during the course of a weight management intervention in women with overweight/obesity was the only variable to consistently predict long-term, sustained weight loss. Findings like these have prompted some scholars to argue that health organizations may benefit from recommending or encouraging flexible dietary control strategies over rigid control strategies for weight and health purposes [83].

Although flexible control may be more adaptive than rigid control, it is important to point out that research has not consistently observed a link between flexible control and positive health outcomes. For example, positive bivariate relationships have been observed between flexible control and adverse health outcomes in some studies of individuals with normal weight and overweight/obesity [11,95,96]. Moreover, flexible control has shown to be highly correlated with rigid control (*r’s* > 0.50), which has not only raised concerns about whether these two restraint components are in fact distinct but may also suggest that encouraging flexible control may unintentionally promote harmful rigid control behaviors [88,95,96]. In fact, more recent work has shown that flexible control is associated with more positive health outcomes only when removing its shared variance with rigid control [11,96], although what these adaptive properties of flexible control are in the absence of rigid control have yet to be determined. For all of these reasons, some have cautioned against the promotion of flexible control strategies for health-related purposes [96].

### 5.2. Self-Monitoring Behaviors

Self-monitoring behaviors are characteristic of eating restraint and are highly prevalent in competitive bodybuilders. Typical self-monitoring behaviors employed by bodybuilders (and the general population) include energy intake tracking, exercise tracking, and body checking (e.g., self-weighing, mirror checking etc.). Some professionals have raised concerns that these self-monitoring behaviors—irrespective of who engages in them—have the potential to precipitate or maintain eating disorder symptoms (e.g., binge eating, rigid dietary control, shape, and weight overvaluation) [100]. In particular, it has been theorized that these self-monitoring behaviors can negatively impact an individual’s self-worth (i.e., where judgements of self-worth become almost exclusively based on weight, shape, and eating, and the ability for one to control them) and induce a pattern of perfectionistic, dichotomous, and obsessive thinking around weight, shape, food, and dieting [8,84].

However, research examining the role of these self-monitoring behaviors on eating disorder symptoms is mixed. For example, several cross-sectional studies have found positive associations between each of these self-monitoring behaviors and a range of eating disorder symptoms (e.g., binge eating, eating concerns, shape/weight overvaluation) in those with and without clinically significant eating disorders, including athletic populations [84,86,100,101,102], and mixed evidence for a potential causal link between shape checking and body dissatisfaction has been found in some experimental studies [103] but not others [104] in non-bodybuilding populations. Prospective studies have also documented positive associations between self-weighing behaviors and binge eating in young adolescent females [105], yet several randomized controlled trials have found no evidence that some monitoring behaviors are associated with adverse outcomes. For example, Steinberg et al. [106] found no significant post-treatment differences in anorectic cognitions, disinhibited eating, binge eating, and depressive symptoms between individuals with overweight/obesity randomized to either a daily self-weighing group or a delayed-intervention control group. Similarly, Jospe et al. [107] found no increase and differences in eating disorder symptoms among individuals with obesity randomized to either a daily weighing intervention, an energy intake tracking mobile intervention, a brief support intervention, hunger training strategies, or a wait-list control.

These mixed findings raise the possibility that these various monitoring behaviors may not be as detrimental as originally thought. As it stands, it is not entirely clear why some of these behaviors may or may not be associated with eating disorder symptoms, as research on moderators (i.e., for whom these relationships exist) and mediators (i.e., how and why monitoring is linked with eating disorder symptoms) is lacking. One hypothesis could be that these monitoring behaviors may be detrimental only when practiced rigidly or inflexibly [87]. Examples of rigid monitoring behaviors could include: repeated weight checking throughout a single day (and a perceived failure if one’s weight-related goals were not achieved); obsessive energy intake counting and exercise tracking for long periods of time; an inability to tolerate not meetings one’s prescribed daily goals (e.g., reacting poorly when eating over a certain energy intake limit or exercising too little). Thus, the extent to which an individual takes an all-or-none approach to self-monitoring could be an important factor influencing any possible relationship between self-monitoring and eating disorder features. Because this has, to our knowledge, not been tested, it is merely a plausible hypothesis that requires empirical exploration.

### 5.3. Eating Based on Internal Hunger and Satiety Cues

The physiological and psychological complications resulting from prolonged energy restriction are well-known [62]. When specific weight or body-composition targets are not immediate goals, a viable alternative to dietary restraint may be intuitive eating. Intuitive eating is a style of eating that is entirely governed by internal hunger and satiety cues [108]. Thus, intuitive eaters (a) eat based on what their body needs rather than on other external or emotional reasons, (b) recognize that all foods serve a variety of important functions (e.g., taste, stamina, performance) depending on the context, and (c) are less preoccupied with food and reject the notion that foods are either “good” or “bad” [108].

The evidence supporting the health benefits of intuitive eating is well-documented. A wealth of cross-sectional research conducted on women and men of many age groups, ethnicities, weight classes, athletic ability, and socioeconomic statuses have reported consistent associations between intuitive eating and positive psychological health outcomes, including lower levels of eating and body image concerns, rigid and flexible dietary control, psychological distress, and psychosocial impairment, and higher levels of positive affect, quality of life and wellbeing, self-compassion, and body appreciation (for a review, see [109]). Further evidence for the potential health benefits of intuitive eating comes from randomized controlled trials examining the impact of interventions grounded in principles of intuitive eating. For example, Bacon et al. [110] compared six months of an intuitive eating-based intervention to a dieting-based weight loss intervention in adult female chronic dieters. The authors found that the group receiving the intuitive eating-based intervention reported significant reductions from baseline to two-year follow-up in total cholesterol, triglycerides, and systolic blood pressure, and also marked improvements in disordered eating behaviors, body image concerns, and depressive symptoms. Importantly, no significant changes in body weight were observed in this group. By contrast, although the weight-loss group reported short-term weight loss, no improvements from baseline to two-year follow-up were observed in key psychological variables (except for disinhibited eating) [110]. Similar findings regarding the beneficial role of intuitive eating-based interventions have been reported elsewhere [111]. Taken together, the available evidence consistently supports the potential health benefits of intuitive eating principles and suggests that this approach to eating may be more adaptive than common forms of eating restraint. However, to our knowledge, no study has examined the potential role and impact of intuitive eating principles in bodybuilding populations, and although evidence upholds the benefits of intuitive eating in a variety of other populations, studying this style of eating in bodybuilders, particularly after their competition, is an important future direction.

## 6. Practical Applications

Given the obligatory need for competitors to reach essential levels of body fat to be competitive, dietary restraint during the competition diet is unavoidable. Additionally, since performance is based on visual judgement of one’s body, monitoring physique progress may also be obligatory, or at least recommended, to ensure the effectiveness of the preparation process. Notwithstanding this, strategies which potentially minimize the potential harm inherent to competitive bodybuilding can still be implemented.

To start, bodybuilding coaches should be aware of the risks involved in competitive physique sport and their own scope of practice. As is standard procedure in the field of personal training [112], a health screening process should occur before a competitor is guided to the stage. It is our recommendation, given high rates of disordered eating and body image dissatisfaction among competitors (e.g., an eating disorder history in 5 of 16 participants reported by Lenzi et al. [14]), that a screening process (questionnaires and/or interview) guided by the input of a registered dietitian and mental health specialist trained in sports dietetics and eating disorders, should respectively occur. This allows at-risk athletes to make an informed decision to undertake competition preparation, and, if made, ensures that the appropriate supportive structures are set in place.

With respect to dietary restraint, a necessary component of bodybuilding that allows one to achieve a specific body composition, we tentatively argue that nutritional strategies and coaching which facilitate flexibility and a “non-dichotomous thinking style” should perhaps be encouraged and implemented as they may result in better mental health without sacrificing bodybuilding performance. As discussed, a flexible approach to dieting has been shown to be healthier (psychologically) and more adaptive than a rigid approach to dieting [83]. In the context of bodybuilding specifically, we remind the reader that some modern-era competitors are adopting the “IIFYM” approach which does not ban specific foods, showing interest exists among athletes for more flexible approaches to nutrition prescription [5,78,79]. Additionally, as discussed in Section 4, the female bodybuilders in an IIFYM group had a better micronutrient profile compared to strict dieters in the sole study comparing macronutrient-based to traditional, strict-dieting practices [80]. Thus, for example, rather than prescribing specific foods in specific amounts at specific times, a macronutrient- or energy content-based plan (that allows for a wide variety of food choices) with flexible meal times and target ranges (e.g., within a plus or minus 10 g or 100 kcal/~400 kj range, respectively) can be implemented after the prescription of basic nutritional education. However, unfortunately, as mentioned in Section 4, while some data indicates adoption of such strategies in certain niches of the bodybuilding community, other data indicate the majority of athletes may not be adopting these arguably more flexible approaches [14,66,70].

Per best-practice guidelines [4], a slower rate of weight loss (e.g., 0.5–1% of body weight per week) should be implemented. This rate is recommended not only because of the mitigated risk of muscle loss but also because faster rates of weight loss are associated with body image concerns and eating disorder symptoms [13]. Slower rates allow higher energy intakes and therefore a more inclusive diet requiring less restraint, with greater energy availability.

The potential physiological benefits of an intermittent dieting approach in athletes, such as lean mass retention [113], enhanced fat loss [114], and maintenance of energy expenditure [114,115], was recently extolled in a speculative review by Peos et al. [116]. While the physiological benefits are worthy of future study, the potential psychological benefits may also be relevant for physique competitors. For example, Wing et al. [117] reported that individuals following a weight-loss plan who were guided to intermittently take weeks off from dieting were able to lose a similar amount of weight as those who dieted continuously for the same period of time. Additionally, more weight loss was maintained at a six month follow up by a group of dieting males who were instructed to intersperse two weeks of eating at maintenance between every two weeks of restriction, compared to a group who dieted for the same length of time continuously [114]. While “diet break weeks” and “refeed days” may help athletes mitigate the negative physiological effects of energy restriction and enhance the process of body composition change, these strategies may also help physique competitors “practice” eating more normally during preparation, thereby facilitating a smoother transition to the offseason with less incidences of binge-eating post competition. While this is a speculative position, readers are directed to the review by Peos et al. [116] as the physiological implications of some intermittent dieting strategies alone may merit their consideration.

Additionally, as per clinical practice for patients with eating and body image disorders [87], assessments of the physique (weight and shape) should occur at structured, semi-regular, and planned intervals, in collaboration with a coach who is aware of and can educate the athlete about the potential harm that could arise from maladaptive body checking (e.g., the tendency for an athlete to focus on very short-term weight fluctuations rather than long-term progress, the degree of scrutiny placed on evaluating one’s own body versus that of other competitors or athletes etc.). In that regard, we recommend advising against frequent (i.e., daily or more than once daily) weight/shape checking; instead, it may be beneficial for the athlete to meet with their coach once per week for a visual assessment of progress. Or, if online, to send pictures or a video of the compulsory poses to the coach for feedback and evaluation. Additionally, while many competitors utilize regular weigh ins (i.e., daily) to assess progress [49,54,79,118], there is potential stress imposed by regular weigh ins [85]. Thus, the athlete could be advised to deemphasize the value put on a singular day’s body weight and create a rolling weekly average in a spreadsheet, only to be assessed at the fortnight’s conclusion.

As discussed previously, binge eating is common following the competition season. Thus, coaches should be diligent and plan a series of strategies for athletes’ post-competition to potentially mitigate this. In case studies, some competitors stay restricted to avoid rapid weight regain post competition [54]. While this may avoid undesirable fat overshooting [119] and negative hormonal adaptations associated with rapid post contest weight regain [52], symptoms of relative energy deficiency may remain months after competition [53] as observed in some competitors continuing to exert dietary restraint post contest [54]. Therefore, we speculate that a middle ground of prescribing a quantified, tapered, but sizeable energy surplus initially to induce steady rather than rapid weight regain, while also providing guidance and coaching to help athletes reconnect with their internal cues of satiety and hunger, may help minimize binge-eating, depression, and any associated negative physiological outcomes.

After this transitional period following a quantitative energy surplus post competition, we recommend athletes adopt a habit-based, qualitatively guided eating approach in the offseason. Rather than weighing and tracking meals and relying on external cues, “modified-intuitive eating” could be implemented. For example, the competitor could maintain certain nutritional habits conducive to offseason bodybuilding progress: An example of this could be including multiple high-protein meals consumed throughout the day (and around training) containing micronutrient dense carbohydrate sources such as dairy, grains, fruits, vegetables, and adequate dietary fat, while letting hunger and satiety cues guide portion sizes. Additionally, this autoregulation of energy could be augmented with bio and performance feedback such as resistance training performance improvements, structured, infrequent physique evaluation, or fortnightly average rate of weight gain (e.g., if average scale weight is increasing too quickly fortnight to fortnight, the athlete could aim to be less full after each meal).

Finally, given the time course to reach competition condition and then fully recover physiologically lasts sometimes 6–9 months or longer [5,53,54], we recommend competitors consider competing at most every other year. Doing so not only allows adequate time for regaining lost muscle mass after contest preparation and gaining new muscle mass in the offseason, but also the return of healthy, normalized eating, guided, at least in part, by internal cues.

Overall, we provide a series of tentative recommendations and future research directions in Table 2 based on the available evidence in both bodybuilding and non-bodybuilding populations. While more work is needed on specific bodybuilding populations, these preliminary recommendations are based on the best available evidence, and we encourage further research. It is our hope that we have generated a series of important hypotheses that sparks a surge in research into this area. Given the abnormally high rates of eating disorders and general psychopathology in this population, we believe that such guidelines are critical to shift the paradigm to healthier and more sustainable practices. While we believe these guidelines should be adopted by coaches and competitors, we encourage research to further enhance the efficacy of our recommendations to establish ever improving guidelines grounded in a larger body of empirical data.

## Figures and Tables

**Figure 1 sports-07-00172-f001:**
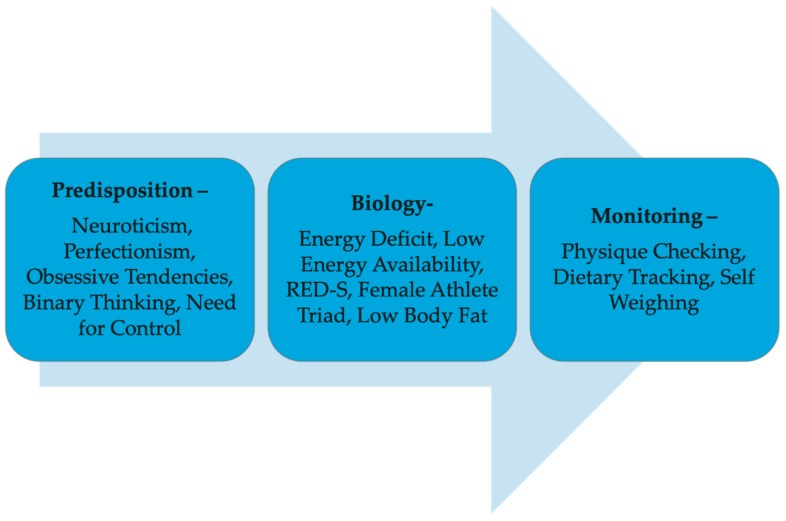
Theoretical drivers of potential harm. Based on the available literature, there are three known potential overarching factors with related sub-factors contributing to the higher risks of body image and eating disorders observed among physique athletes: (1) a predisposition towards these disorders, related to certain personality traits; (2) the biological response to energy restriction and low energy availability, potentially resulting in Relative Energy Deficiency in Sport (RED-S) and the female athlete triad, further exacerbated by low body fat; and (3) the monitoring behaviors used to exert the restraint necessary to follow and ensure the effectiveness of competition preparation diets.

**Table 1 sports-07-00172-t001:** Key studies of the biological and psychological effects of physique contest preparation.

Reference	Participants	Biological Effects	Psychological Effects
Rossow et al. (2013)	N = 1 Male, natural bodybuilder	- Blood pressure, testosterone, T3, and T4 decreased but recovered quickly - Ghrelin and cortisol levels increased	- Mood disturbance
Kleiner et al. (1990)	N = 27 (70% male)	- Plasma glucose and magnesium decreased - Hemoglobin levels increased - Menstrual irregularities (25% of women)	
Kistler et al. (2014)	N = 1 Male, natural bodybuilder	- Bone mineral content and density increased - Blood pressure and arterial stiffness decreased	
Trexler et al. (2017)	N = 15 (47% male)	- Cortisol increased but recovered - Testosterone recovered later - Ghrelin increased - Leptin increased - Insulin unchanged	
Hulmi et al. (2017)	N = 50 (all female)	- Leptin, T3, testosterone, estrogen, bone mass decreased but mostly recovered - Heart rate decreased - Menstrual irregularities	- Mood changes (decreased vigor) but recovered
Halliday et al. (2016)	N = 1 Female, natural bodybuilder	- Energy availability decreased - Loss of menses (recovered 71 weeks post-competition)	
Andersen et al. (1994)	N = 49 (all males)		- Binge eating frequency - Food preoccupation - Depression, anxiety, anger
Walberg-Rankin et al. (1993)	N = 6 (all female)	- Vitamin deficiency - Menstrual irregularities - Estradiol and progesterone decreased but recovered	
Robinson et al. (2015)	N = 1 Male	- Resting metabolic rate decreased	- No mood changes - No binge eating
Rohrig et al. (2017)	N = 1 Female	- Leptin and thyroid stimulating hormone decreased - Estradiol and cortisol unchanged	- Energy decreased - Tiredness increased

**Table 2 sports-07-00172-t002:** Hypothetical seasonal guidelines for mitigating harm in physique sport.

Competition Preparation	Non-dichotomous “good or bad” or “on or off” view of foods or diet. Moderate weight loss rates. Macronutrient or energy intake ranges versus food source restriction. Flexible meal schedules. Scheduled, supervised physique assessment, body weight averages. Intermittent dieting strategies; semi-regular diet break weeks or days.
Transition to Offseason	Quantified, tapered, but substantial energy surplus. Return to offseason body composition in ~1–2 months. Intentional re-introduction of and focus on internal cues; e.g., hunger.
Offseason	Adoption of primarily internal-cue guided energy intake. Maintain sport-supportive nutrition habits versus relying on tracking. Biofeedback (e.g., rate of weight gain) to augment energy intake. Qualitative response to biofeedback, e.g., “be less full after meals”.

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
