# Peer review of "Towards a Sustainable Nutrition Paradigm in Physique Sport: A Narrative Review"

_sports, 2019, doi:10.3390/sports7070172_

Round 1

Reviewer 1 Report

Towards a Sustainable Nutrition Paradigm in Physique Sport

General

Overall, this is a well written review with a novel research question, and one of the few review for bodybuilding athletes. I would like to commend the authors for the effort and the updates on this sport. The authors establish a threefold purpose during this review related to the mental health of bodybuilder athletes and mainly aim to answer 1) pre-existing, or a predisposition to develop, body image or eating disorders; 2) biological effects of energy restriction on eating psychology and 3) their dietary restraint attitude, and resultant physique, exercise and nutrition monitoring behaviour.

I compliment the author’s approach for the well-established study both scientific and experimental. However, there are specific points that I would like to be clearer inside the manuscript.

Abstract:

Line 2: … have higher rates of body image and eating disorders… I understand the eating disorders, but I thing that authors should make clear that increased lean mass and not just body image is a main goal of training for these athletes.

Introduction:

Line 23: As much as I want to agree with this statement, it is not correct. In all bodybuilding categories the most important factor is the low percentage of body fat and the high muscular definition. Authors should clarify that inside the manuscript.

Line 40: Now in every review manuscript readers should know the following: 1) the description of the key words authors used in order to describe predisposition, biology and monitoring 2) how many studies did the authors found during the searching 3) how many were relative and how many were excluded and 4) the main criteria of the presented studies inside the manuscript. 

I recommend to authors to include inside the manuscript a paragraph with the above information’s.

2. Psychological profiles and tendencies  

I agree with the profiles and tendencies presented here. However, what can the authors suggest to athletes, who are using illegal substances, in order to improve their psychological profile (irritation, self-esteem, inferiority syndrome etc.). Are there any research evidence to overcome these psychological factors in these athletes like updates in these matters or meetings with special psychologists?

Line 50: This is not what differentiates bodybuilders from other athletes this is the main purpose and their goal during training.

Line 54: Delete …thus…

Line 72: Change …and… to …while…

Line 89: Please replace everywhere inside the manuscript to…  et al.,

3. Demands and effects of contest preparation

I am not quite sure that the title here is correct. Inside the following 4 paragraphs authors present evidence from the post-competition period following binge-eating. So I recommend to authors to re-write the title of the paragraph.

Line 140: It is not only the addition of the aerobic exercise, but also the addition of resistance training with multijoin exercises for the whole body, small rest periods and many repetitions. Please consider this.

Line 179:

Are these two studies referred to bodybuilders?

Dulloo, A.G., J. Jacquet, and L. Girardier, Poststarvation hyperphagia and body fat overshooting in humans: a role for feedback signals from lean and fat tissues. Am J Clin Nutr, 1997. 65(3): p. 717-23. 609

 Weyer, C., et al., Energy expenditure, fat oxidation, and body weight regulation: a study of metabolic adaptation to long-term weight change. J Clin Endocrinol Metab, 2000. 85(3): p. 1087-94.

Line 203: This is the conclusion of Suffolk et al., So I wonder, what is the conclusion of the authors according to contest preparation??? This paragraph needs to be re-written and aim to the pre contest.

4. Nutrition and physique management practices of physique competitors

I suggest to authors to separate the meanings of ‘physique competitors’ and ‘bodybuilder competitors’. If you refer to physique athletes there are many differences from bodybuilding athletes both in males and females. Please clarify. Also, when authors refer to physique management inside the title, what do they refer to? Is this a nutritional management?

Line 223: Please replace prep to preparation inside the manuscript.

Line 226: Delete …even…

5. Nutritional approaches for minimizing harm

According to authors no evidence exists for bodybuilders but only from general population. Then how can authors recommend these results to athletes whose main purpose is the increase of lean mass and the reduction of fat?

Line 273: Please re-phrase the sentence.

Line 277: 5.1. Dietary restraint: Distinguishing between flexible and rigid control. Please change the order of flexible and rigid inside the title in order to queer inside the manuscript.

Line 288: Is rigid approach dieting related to the psychological profile of bodybuilders both males and females?

Line 339 and 344: Is there a possible link between these statements and bodybuilding or physique?

Line 364: 5.3. Eating based on internal hunger and satiety cues: Is there a link between internal hunger and bodybuilding or physique?

6. Practical applications

Line 412: Indeed authors discussed it but, no link was observed with bodybuilding performance. So how can authors claim that?

Line 452: Please change …clients… to ….athletes...

Line 481: Now this is confusing. If your recommendations needs further research, then why should the readers of this paper follow them?

Reviewer 2 Report

This is a well-written, interesting and helpful review. Concerns are listed below.

MAJOR

1.     The manuscript needs a dedicated section detailing the evidence that these athletes are at high risk of psychopathology related to food and body image. There is an underlying assumption in the paper that the reader knows this, but not every reader does.

2.     Similar to the above, the manuscript needs some dedicated text outlining how this psychopathology related to food and body image is perceived by the athletes themselves as being disruptive to their wellbeing. While it is clear that the athletes engage in some behaviors that are considered pathological in other contexts (e.g. rigid control of food and beverage intake, frequent body checking), are these behaviors not serving the athletes well? Do they see it as a problem themselves? Is it something that they want to change?

3.     The Authors need to avoid any implications in the paper that the sport leads to the psychopathologies mentioned. In several places in the text, this has been suggested – either explicitly or implicitly. For example, lines 128-129, among other examples. In fact, it is not known which is the ‘chicken’ and which is the ‘egg’. Do people with a tendency to this psychopathology become attracted to the sport, or does the sport induce the psychopathology, or both? This is not known, and some more frequent reminders of this in the text will be helpful.

4.     Similar to the above, around line 339 a suggestion of causality between participation in the sport and psychopathology is given. It would be helpful to give more information about the study that led to that suggestion, so that the reader can more clearly assess whether causality is plausible.

5.     At around line 64, it is stated that “coaches and athletes should ensure that this pursuit does not become pathological and impact a bodybuilder’s sense of self-worth, esteem, or identity outside of the competitive confines of his or her sport.” It would be helpful at this point to give some suggestion as to how coaches and athletes can do this, or at least refer the reader to a later section of the manuscript where this will be dealt with, or refer the reader to other literature.

6.     The manuscript needs to use people-first language. For example, instead of “overweight/obese women”, it is better to say “women with overweight or obesity”.

7.     Using metric rather than imperial terms would be helpful. For example, “kilojoule tracking” rather than “calorie tracking” (or use both).

8.     The Authors “argue that nutritional strategies and coaching which facilitate flexibility and a “non-dichotomous thinking style” should be encouraged and implemented.” However, what is the evidence that this leads to the level of weight loss and body composition changes that these athletes require to be competitive in the sport? If there is no evidence for this, then it should be stated clearly, so that readers know this is a hypothesis that requires further testing (e.g. in RCTs).

9.     At lines 413-417, a strategy is suggested which appears to be similar to what athletes are already doing, as mentioned earlier in the text. How does this proposed strategy differ from what athletes already do (or what coaches recommend), in general?

10. Table 1 is a hypothesis for further testing. This should be stated explicitly also in the table title, not just in the text, so that readers know that this table is not evidence-based guidelines, but rather it is recommendations for further testing that could cautiously be applied in real life to see how it works. For example, in the obesity space, it was shown in papers involving Byrne S and Fairburn C that implementing an intervention to reduce dichotomous thinking had no impact on outcomes from a weight management program.

Reviewer 3 Report

The manuscript is a narrative review around some of the issues and complications physique athletes face.  The review addressed both the biological and psychological issues in these athletes and gives thoughtful recommendations at the end of the article.  This is a very interesting topic, and a thorough narrative review, that I believe is warranted.

General Comments

Biological effects of energy restriction are mentioned, but only a small proportion of the biological effects are mentioned (T3 and estradiol).  There are many other effects of an energy deficit including changes in metabolism ( demonstrated by decreases in resting energy expenditure in addition to the changes thyroid hormones).  Women with an energy deficit have also demonstrated negative impacts on bone health and reproductive health (The Female Athlete Triad literature has a lot of strong literature that could improve the biological aspects of this review).  It may help to separate out the psychological and biological sections of your review. 

To help separate out all the information that you are trying to present.  I recommend adding a table that includes the major studies you discuss, the type of population they researched and the specific findings (i.e. high cognitive restraint or low resting metabolic rate).  Overall the writing and research is excellent, I just want to try and help you organize the information to make it easier for the reader.

Minor Comments.

Title

Be clear that this is a narrative review in the title.

Abstract

Be clear that this is a narrative review in the abstract.

Add a final sentence to your abstract about what you concluded or what you will want the reader to take home.

Figure 1.

 Please include a figure legend that describes the figure in detail.  Also please expand the biological impacts in this figure.

Body

Pg.3 line 48 – “higher rate of muscle dysmorphia” – Be more specific.  Give the reader some numbers.

Pg.4 line 120-125 – Compare these rates of eating disorders to the regular population and to other sports.

Round 2

Reviewer 1 Report

No comments.

Reviewer 2 Report

This review is very interesting and well presented. All my concerns in peer review are addressed in the revised manuscript.